# Improved Fluorescent Proteins for Dual-Colour Post-Embedding CLEM

**DOI:** 10.3390/cells11071077

**Published:** 2022-03-23

**Authors:** Dingming Peng, Na Li, Wenting He, Kim Ryun Drasbek, Tao Xu, Mingshu Zhang, Pingyong Xu

**Affiliations:** 1Key Laboratory of RNA Biology, Institute of Biophysics, Chinese Academy of Sciences, Beijing 100101, China; dmpeng7@gmail.com (D.P.); nali9415001@gmail.com (N.L.); hewenting@ibp.ac.cn (W.H.); 2Bioland Laboratory (Guangzhou Regenerative Medicine and Health Guangdong Laboratory), Guangzhou 510005, China; 3Sino-Danish College, University of Chinese Academy of Sciences, Beijing 100101, China; 4Sino-Danish Center for Education and Research, Beijing 100039, China; ryun@cfin.au.dk; 5Center of Functionally Integrative Neuroscience, Department of Clinical Medicine, Aarhus University, 8000 Aarhus, Denmark; 6National Laboratory of Biomacromolecules, CAS Center for Excellence in Biomacromolecules, Institute of Biophysics, Chinese Academy of Sciences, Beijing 100101, China; 7College of Life Sciences, University of Chinese Academy of Sciences, Beijing 101408, China

**Keywords:** dual-colour CLEM, high SBR, RSFP, probe development

## Abstract

Post-embedding correlative light and electron microscopy (CLEM) has the advantage of high-precision registration and enables light and electron microscopy imaging of the same slice. However, its broad application has been hampered by the limited available fluorescent proteins (FPs) and a low signal-to-background ratio (SBR). Here, we developed a green photoswitchable FP, mEosEM-E with substantially high on/off contrast in EM samples embedded in Epon resin, which maximally preserves cellular structures but quenches the fluorescence of FPs. Taking advantage of the photoswitching property of mEosEM-E, the autofluorescence background from the resin was significantly reduced by a subtraction-based CLEM (sCLEM) method. Meanwhile, we identified a red fluorescent protein (RFP) mScarlet-H that exhibited higher brightness and SBR in resin than previously reported RFPs. With mEosEM-E and mScarlet-H, dual-colour post-Epon-embedding CLEM images with high SBR and no cross-talk signal were successfully performed to reveal the organization of nucleolar proteins. Moreover, a dissection of the influences of different EM sample preparation steps on the fluorescence preservation for several RFPs provides useful guidance for further probe development.

## 1. Introduction

Light microscopy (LM) highlights and discriminates one or a few biomolecules using fluorescent labelling. Electron microscopy (EM), on the other hand, reveals the ultrastructural context of the cell where the biomolecules reside. Correlative light and electron microscopy (CLEM) [1] integrates the information of both LM and EM in a complementary way, and thus provides localization, structural and functional insights into the biomolecular machines, and shows great application prospects in cellular physiology [2], virology [3] and neuroscience [4]. In CLEM imaging, LM can not only indicate the location of the target protein, but also be used as a guide to quickly find slices containing target cells or proteins in a large number of EM slices, and to quickly locate the cell on the target slice for EM imaging. The advantage of fluorescence navigation is unique and is not possessed by EM labelling techniques, such as immunogold labelling, APEX-gold [5], APEX2 [6], miniSOG [7], and other metal [8] or chemical [9] tags.

The same as any other imaging technique, the practical performance of CLEM is largely determined by the labelling probe. However, most fluorescent proteins (FPs) experienced severe fluorescence loss during standard EM sample preparation, making it difficult to correlate with high precision signals from LM and EM on the same section. To circumvent this problem, several strategies have been used. Pre-embedding CLEM [10,11] obtains fluorescence microscopy (FM) images ahead of the EM sample preparation, avoiding fluorescence quenching caused by a series of chemical treatments. However, because of the morphology distortion during EM sample preparation and sectioning, it suffers from poor registration between the FM and EM images. Several modified EM sample preparation protocols with no or reduced OsO_4_ concentration have been reported to preserve sufficient fluorescence signal for post-embedding CLEM [12,13], however, deteriorated EM images were often observed.

In 2015, Paez-Segala et al. engineered the first FP, mEos4b, to retain fluorescence after 1% OsO_4_ treatment [14]. However, the hydrophilic GMA resin was used for mEos4b-labelled samples. Compared to the GMA resins, the hydrophobic Epon resin has advantages in maintaining cellular ultrastructure as well as sectioning quality [3,15] due to its higher toughness and hardness. Nevertheless, the Epon resin undermines fluorescence more severely. To solve this problem, we have previously developed mEosEM [16], an FP that survives 1% OsO_4_ fixation and Epon embedding and enables super-resolution CLEM (SR-CLEM) due to the preserved photomodulable property. Recently, Tanida et al. found that mKate2 [17], mCherry2, mWasabi, and GoGFP-v0 [18] could also preserve fluorescence after standard EM sample preparation and dual-colour CLEM imaging was achieved using mWasabi and mCherry2. However, both the green and the red channel images showed very low signal-to-background ratio (SBR), making it difficult to distinguish the real fluorescence signals from that of the background or noise.

Low SBR can be attributed to three aspects. First, most of the fluorescence signals of the FPs mentioned above are quenched after transmission electron microscopy (TEM) sample preparation, and only a small amount of the remaining fluorescence signal is used for CLEM imaging. Second, the thickness of ultrathin sections is generally only about 100 nm, and the number of FP-labelled molecules in the sections is limited. Third, the Epon resin has strong autofluorescence, especially in the green channel (Appendix A). Therefore, developing FPs with high in-resin SBR and repressing the autofluorescence background are effective ways to solve the problem. For dual-colour CLEM, another phenomenon worth noting is that red fluorescent proteins (RFPs) on Epon-embedded slices emitted green fluorescence when illuminated with 488-nm laser (Appendix A), which interfered with the signal in the green channel. Therefore, FPs and imaging methods that can solve these problems are desirable.

Reversibly photoswitchable FP (RSFP) can be utilized to suppress the unmodulatable fluorescent background and enhance the signal contrast by means of optical lock-in detection (OLID) [19], synchronously amplified fluorescence image recovery (SAFIRe) [20,21] and out-of-phase imaging after optical modulation (OPIOM) [22]. We speculated that similar strategies could be applied with an OsO_4_—and Epon-resistant RSFP to eliminate the resin background and the RFP crosstalk signal. In this study, we developed a fluorescence background-reduced CLEM method (sCLEM) using a simple subtraction of images of RSFPs at fluorescent on- and off-states and efficiently extracted the fluorescence signals of the FPs from that of the background. The higher on/off contrast of the RSFP, the better SBR of the final image. Therefore, we evolved a mEosEM variant termed mEosEM-E, with high on-state brightness and on/off contrast ratio after standard EM sample preparation and demonstrated its utility in sCLEM imaging.

On the other hand, despite a few FPs were reported for post-Epon-embedding CLEM [14,16,17,18], all these probes were discovered by the use of an OsO_4_ resistance assay, the results of which are sometimes inconsistent with the final performance of the probe in CLEM imaging. As a matter of fact, nearly every step during EM sample preparation will quench the fluorescence of FPs. However, a comparative study that dissects the influence of each step on fluorescence signal reduction is lacking. To develop an optimal RFP that can be coupled with mEosEM-E for dual-colour CLEM imaging, we assessed the fluorescence preservation of nine commonly used RFPs after each key step of EM sample preparation, including pre-fixation, OsO_4_ fixation, ethanol dehydration, and Epon embedding. The results showed that the OsO_4_ resistance assay is not the optimal criterion for CLEM probe development, while the performance of a probe should be evaluated in the final sample section. Using this approach, we found that mScarlet-H exhibited higher brightness and SBR in the Epon resin compared with previously reported RFPs and is suitable for same section post-Epon-embedding CLEM. Finally, high SBR and accurate dual-colour CLEM imaging of nucleolar proteins was successfully achieved for the first time using mEosEM-E and mScarlet-H double labelling.

## 2. Materials and Methods

### 2.1. Development of mEosEM-E

Saturation mutagenesis of mEosEM at His63 was performed in the pEGFP-N1-mito-mEosEM plasmid with Q5 high fidelity polymerase (New England Biolabs, Ipswich, MA, USA). The amplified fragments containing homologous arms and mutation sites were transformed into the Top10 competent cells (Tsingke, Beijing, China) and sequenced (Tsingke, Beijing, China). The HEK 293T cells were transfected with sequencing validated mutants and were prepared following the standard EM sample preparation procedure. Cell slices with a thickness of 100 nm were imaged under a homemade wide field fluorescence microscope (IX71 body and PLAN APO 100×, 1.49 NA oil objective, Olympus, Tokyo, Japan) to characterize the photoswitching properties of different mutants. The mEosEM-E (mEosEM H63E) was identified for its highest photoswitching contrast ratio.

### 2.2. Plasmid Construction

For prokaryotic expression plasmids pRsetA-RFPs (mScarlet, mScarlet-I, mScarlet-H, mKate2, FusionRed-MQV, mCherry2, mRuby3, mApple, tdTomato), RFPs fragments were PCR amplified and digested with BamHI and EcoRI restriction enzymes. Then fragments were ligated into the pRsetA vector digested with the same enzymes. For eukaryotic expression plasmids, pmEosEM-N1-mito was digested with AgeI and NotI restriction enzymes to replace mEosEM with mEosEM-E or RFPs fragments with the same restriction enzyme sticky ends. The pEGFP-C1-Sec61, pmEos3.2-N1-H2B, and pmEosEM-C1-LaminA plasmids were digested with AgeI/BglII, XhoI/NotI, and NheI/BglII restriction enzymes, respectively, to replace EGFP, mEos3.2 and mEosEM with the mScarlet-H fragment. For the construction of pmEosEM-C1-B23, pmEosEM-C1-Nop52, and pmScarlet-H-C1-Nopp140 plasmids, the full-length cDNA of B23, Nop52 and Nopp140 were PCR amplified from the HEK 293T cDNA library, digested with EcoRI/SalI, HindIII/SalI, and BglII/SalI restriction enzymes and inserted into pEGFP-C1-mEosEM-LaminA and pEGFP-C1-mScarlet-H-LaminA plasmids to replace LaminA. The Q5 polymerase and T4 ligase were purchased from New England Biolabs (Ipswich, MA, USA). All restriction enzymes were purchased from Thermo Fisher Scientific (Waltham, MA, USA).

### 2.3. Cell Culture and Transfection

The HEK 293T and U-2 OS cells were cultured in Dulbecco’s Modified Eagle Medium (Gibco, Thermo Fisher Scientific, Waltham, MA, USA) and McCoy’s 5A Modified Medium (Gibco, Thermo Fisher Scientific, Waltham, MA, USA), respectively, supplemented with 10% FBS (Gibco, Thermo Fisher Scientific, Waltham, MA, USA) and 1% penicillin-streptomycin (TransGene Biotech, Beijing, China). Cells were maintained at 37 °C in an incubator supplied with 5% CO_2_ (vol/vol). Transfections were performed with purified plasmids using lipofectamine 2000 (Invitrogen, Waltham, MA, USA) (with a ratio of 1 µg DNA: 3 µL lipofectamine) in 6 cm petri dishes or 12-well cell culture plates following the manufacturer’s instructions.

### 2.4. Photoswitching Property Analysis

The HEK 293T cells expressing mitochondria-targeted mEosEM and mutants were used for photoswitching property analysis. For the analysis before EM sample preparation, cells were excited by continuous 488-nm laser (0.41 kW/cm^2^), while every 10 s, a 405-nm laser (0.21 kW/cm^2^) pulse of 0.1 s were added to turn on the FPs. Six cycles were recorded for all FPs. For analysis after EM sample preparation, 100 nm cell sections were imaged under a continuous 488-nm laser for 50 frames, after which the 405-nm laser was added for 1 s to record the fluorescence signal of the FPs at the on-state. The contrast ratio was calculated as follow:
Contrast ratio = (Max − Mean)/Mean
where Max represents the maximum value of the signal in all acquired images, Mean represents the averaged signal value of the first 20 frames acquired before the application of 405-nm laser.

### 2.5. High-Content Analysis of Pre-Fixation, Post-Fixation, and Dehydration Resistance of RFPs

The U-2 OS cells expressing pRFPs-mito were seeded into 96-well optical polymer base microplates and fixed with 2% paraformaldehyde (Electron Microscopy Sciences, Hatfield, PA, USA) and 2.5% glutaraldehyde (Electron Microscopy Sciences, Hatfield, PA, USA) in 100 mM PBS at 37 °C for 15 min. The pre-fixed cells were washed 4 times with PBS and then post fixed with 1% OsO_4_. After 10 min incubation at 4 °C, OsO_4_ was removed and cells were washed 5 times with PBS and stored in PBS. For dehydration, PBS buffer was replaced with absolute ethanol for 20 min without washing. Fluorescence images were acquired by the Opera Phenix™ High Content Screening System (PerkinElmer, Waltham, MA, USA) using a 20×, 0.4 NA water objective with an excitation laser of 568 nm. Data quantification and analysis were performed using Harmony 4.9 software (PerkinElmer Waltham, MA, USA).

### 2.6. Protein Purification and Thermostability Measurement

The BL21(DE3) competent cells (Tsingke, Beijing, China) were transformed with the prokaryotic expression plasmids pRsetA-RFPs and single clones were cultured in liquid LB medium to the logarithmic growth phase. Then 0.8 mM IPTG (Isopropyl β- d-1-thiogalactopyranoside) was added to induce the expression of the RFPs. After induction at 16 °C for 24 h, cells were harvested by centrifugation. The pelleted bacteria were resuspended in binding buffer including 10 mM imidazole and lysed by ultrasonication. Protein was purified through affinity chromatography (Ni-NTA His-Bind resin, Qiagen, Hilden, Germany), followed by gel filtration chromatography (Superdex 200 Increase 10/300 GL, GE Healthcare, Chicago, IL, USA). Purified proteins were diluted in PBS (pH = 7.2) and the fluorescence intensity was recorded at 60 °C in the Rotor-Gene 6600 real-time PCR cycler (Qiagen, Hilden, Germany) for 18 h. The thermostability was defined as the ratio between the initial and the final fluorescence intensities.

### 2.7. Photostability Measurement

The HEK 293T cell samples expressing mitochondria-targeted RFPs were prepared under standard EM sample preparation procedure and sectioned into 100 nm slices. Sample slices were illuminated with a 561-nm laser and the fluorescence signal was acquired by time-lapse imaging. Photostability was defined as the time when fluorescence intensity reached 1/e of its initial.

### 2.8. EM Sample Preparation

Successfully transfected cells were trypsinized and then harvested by centrifugation. Cells were fixed with 2% paraformaldehyde (Electron Microscopy Sciences, Hatfield, PA, USA) and 2.5% glutaraldehyde (Electron Microscopy Sciences, Hatfield, PA, USA) in 100 mM PBS at 4 °C overnight. Pre-fixed cells were washed 2 times with PBS and 2 times with Milli-Q water on ice. Then post-fixated with 1% OsO_4_ on ice for 1 h. After 4 washes with Milli-Q water, cells were gradually dehydrated with a series of concentration gradients of ethanol (30% 50% 70% 90% 100%) for 6 min each and followed by dehydration with 100% acetone for 6 min. Cells were then infiltrated step by step in mixtures of Epon (50% Epon812, 30.5% NMA, 18% DDSA, 1.5% DMP-30) and acetone with gradient concentrations (50% Epon for 2 h, 75% Epon for 3 h, 100% Epon for 12 h, 100% Epon for 12 h). Finally, cells were embedded in 100% Epon at 60 °C for 16 h and then sectioned into 100 nm slices for subsequent experiments.

### 2.9. CLEM Imaging

Cleaning and coating of the coverslips with pioloform films (Ted Pella, Redding, CA, USA) were processed as previously reported [23]. Sectioned slices were placed on well prepared coverslips and submerged with mowiol buffer to recover the fluorescence. After a 30 min incubation, slices were imaged using a widefield fluorescence microscope (Olympus IX71, Tokyo, Japan) equipped with a 100×, 1.49 NA oil objective (Olympus PLAN APO, Olympus, Tokyo, Japan). Images were acquired using an electron-multiplying charge-coupled device (EMCCD) camera (Andor iXon DV-897 BV, Belfast, UK) with a detection gain of 300. For the red channel, 100 frames were acquired for averaging during excitation by a 561-nm laser. For the green channel, 20 frames were acquired first using a 488-nm laser excitation alone, then another 20 frames were acquired while a 405-nm laser pulse of 1 s was added to switch on the mEosEM-E molecules. For both channels, the exposure time was 50 ms. After fluorescence signal recording, DIC images of 100×, 16×, and 10× magnifications were sequentially collected to assist the target retrieving during subsequent EM imaging. After LM imaging, a rectangle on the pioloform film with a slice on it was scored by a knife. 12% hydrofluoric acid was dropped on the periphery of the rectangle to detach the pioloform film from the glass coverslip. When the coverslip was submerged under water, the detached pioloform film would float on the surface and was captured by an uncoated slot grid. Next, the section slice was stained with 2% UA and 1% Sato’s triple lead. Finally, the sample was imaged under TECNAI SPIRIT TEM (FEI, Hillsboro, OR, USA). Gold nanoparticles were used as the fiducial marker. The FM and EM images were correlated by eC-CLEM following the previous protocol [16].

### 2.10. Registration of Green and Red Channel FM Images

TetraSpeck™ Microspheres (Thermo Fisher Scientific, Waltham, MA, USA) were diluted in PBS buffer and spotted on a coverslip (Fisher Scientific, Hampton, VA, USA). Dual-colour fluorescence signals were recorded simultaneously under 488- and 561-nm lasers. The registration was performed with the Fiji plugin “Multi Registration” in ImageJ (Version 1.8.0_172, NIH, Bethesda, MD, USA).

## 3. Results

### 3.1. Green mEosEM-E with High SBR for sCLEM

To obtain a CLEM probe with high on/off contrast, we chose mEosEM as the template. The mEosEM is a photoconvertible FP (PCFP) that can be converted to an RFP upon 405-nm illumination, but it can also photoswitch in the green state like an RSFP. After the TEM sample preparation, mEosEM lost the characteristic of light conversion while retaining the photoswitching property [16]. However, this photoswitching property was not used for background removal in CLEM imaging, and the on/off contrast was not optimized for this use.

Our previous study revealed that the first amino acid of the chromophore tripeptide (XYG) of the Eos FP family not only determines the photomodulable type (RSFP or PCFP) of the FP but also greatly affect its photoswitching property [24]. We reasoned that because the first amino acid of the chromophore is located inside the barrel structure of the FP, mutagenesis at this site would have no negative effect on the resistance to the EM sample preparation, while largely tuning the photoswitching contrast. As expected, saturation mutagenesis at His63 of mEosEM produced a series of RSFP mutants displaying a variety of switching contrasts and kinetics (Appendix A, Appendix A). We found that among several improved mutants, mEosEM-E (mEosEM H63E) displayed substantially improved contrast, in other words, largely reduced normalized residual fluorescence as compared to mEosEM after TEM sample preparation (Figure 1A). The on/off contrast ratio of mEosEM-E is ~2.7-fold higher than that of mEosEM on average (Figure 1B, Appendix A).

To demonstrate the superiority of mEosEM-E in CLEM, we imaged HEK 293T cell sections expressing mitochondria-targeted mEosEM-E or mEosEM. Fluorescence image sequences were acquired under the illumination of the 488-nm laser alone followed by the 488- and 405-nm lasers together. We termed the images acquired without the illumination of 405-nm laser as “OFF images” and those with 405-nm laser as “ON images”. Every 20 frames of the OFF and subsequent ON images from the image stack were averaged respectively. Then, the subtraction was performed pixel by pixel between the averaged ON and OFF images to acquire the final “ON-OFF image” (Appendix A). We named the subtraction-based LM as sLM, and accordingly CLEM as sCLEM. It can be clearly seen that the auto-fluorescent background in the OFF image was almost completely removed in the ON-OFF image of mEosEM- and mEosEM-E-labelled cells (Figure 1C). For both mEosEM and mEosEM-E, the SBR in the ON-OFF image was significantly increased compared to the ON image (Figure 1D,E, Appendix A and Appendix A). Notably, the SBR of mEosEM-E fluorescence is 3.46-fold higher than that of mEosEM in the ON-OFF image (Figure 1F, Appendix A). Further exploration revealed that the substantial higher SBR in the ON-OFF image of mEosEM-E is mainly attributed to its higher SBR in the ON image (Appendix A, Appendix A and Appendix A).

Next, same section post-Epon-embedding sCLEM was demonstrated in HEK 293T cells, of which the nuclear lamina (Figure 1G–I) and the mitochondria matrix (Figure 1J–L) were labelled by mEosEM-E, separately. The results showed that high SBR and nearly background-free fluorescence images were obtained and well aligned with the EM images.

Altogether, mEosEM-E proved to be a better CLEM probe for high SBR imaging and is potentially useful for investigating proteins of low expression level.

### 3.2. Identification of mScarlet-H as a High-Performance Red CLEM Probe

To find the optimal red probe for dual-colour CLEM, nine commonly used RFPs were chosen for investigation: mScarlet, mScarlet-H, mScarlet-I, FusionRed-MQV, mRuby3, mApple, tdTomato, mKate2 [17], mCherry2 [18], of which the latter two were previously demonstrated feasible for post-Epon-embedding CLEM. Instead of performing only the OsO_4_-resistant assay reported in the previous papers [16,17,18], we thoroughly analysed the fluorescence preservation after each key step of the EM sample preparation procedure, including aldehyde fixation, OsO_4_ fixation, ethanol dehydration, Epon embedding, and high-temperature polymerization.

A direct comparison of the absolute fluorescence intensity in fixed cells, which is a representative of the practical performance of the probe, showed that pre-fixation with 2% formaldehyde and 2.5% glutaraldehyde already significantly reduced fluorescence intensities of several RFPs, up to 50% of their initial value. While a few others, such as mScarlet-H and FusionRed-MQV were much less affected by pre-fixation, indicating a variated response among different RFPs (Appendix A, Appendix A and Appendix A). The fluorescence intensities of all RFPs were markedly reduced after 1% OsO_4_ fixation, nevertheless, mScarlet-I, mRuby3, mScarlet, and mScarlet-H are the top four FPs retaining high residue fluorescence, which are all substantially higher than that of the previously reported mKate2 and mCherry2 (Figure 2A, Appendix A). However, the subsequent dehydration step using 100% ethyl alcohol changed the ordering of RFPs, making only mRuby3 standout, while others diminished to a similar level (Figure 2B, Appendix A). We speculated that the advantage of mRuby3 at this stage is due to the insensitivity of mRuby3 to the dehydration treatment, which was later confirmed by the results from dehydration treatment without OsO_4_ fixation (Appendix A, Appendix A). Notably, the robustness of mRuby3 to the dehydration treatment following 1% OsO_4_ fixation is not as strong as that to a single dehydration treatment, indicating a superimposed effect of the two treatments.

Next, we performed an in vitro thermostability test at 60 °C to mimic the effect of high temperature on fluorescence preservation during Epon resin polymerization. Among the nine selected RFPs, mScarlet-H showed the best thermostability that is significantly higher compared to the others (Figure 2C, Appendix A). After complete EM sample preparation, mScarlet-H and mScarlet-I both showed the highest residue fluorescence, which is significantly higher than those of previously reported mKate2 and mCherry2, while the residue fluorescence of mRuby3 was surprisingly low (Figure 2D, Appendix A). Statistic results showed that mScarlet-H and mScarlet-I have the highest SBR among the tested RFPs (Appendix A, Appendix A). We speculated that mRuby3 might be super sensitive to the Epon embedding procedure. In addition, even though mApple and tdTomato show comparable fluorescence residue to mKate2 and mCherry2 after OsO_4_ fixation and dehydration treatment, and even better thermostability than mCherry2 and mKate2, they totally lose their fluorescence after complete EM sample preparation (data not shown).

Additionally, we also measured the photostability of these RFPs on standard EM sample sections under the illumination condition of CLEM imaging. Here, mScarlet-H had higher photostability than the other RFPs except mRuby3 (Figure 2E,F, Appendix A).

Taken together, we recommend mScarlet-H as the best red probe for standard CLEM as it has the highest in-resin fluorescence and a preferable photostability among RFPs after complete TEM sample preparation.

### 3.3. mScarlet-H Is a Generalizable Red Probe for Post-Epon-Embedding CLEM

To exemplify the utility of mScarlet-H in Epon-embedded same section CLEM, we constructed mScarlet-H fusions with the mitochondria-targeting peptide, the endoplasmic reticulum membrane protein Sec61β, and histone H2B, individually. Cell sections transiently expressing mScarlet-H fusions were prepared according to the standard TEM sample preparation procedure. The fluorescence signal of mScarlet-H from thin sections (100 nm) was consecutively recorded for multiple frames. The first 100 frames were averaged to smooth the background noise and increase the signal-to-noise ratio. Mitochondria, endoplasmic reticulum, and Histone H2B were correctly targeted and clearly identified under fluorescence microscopy (FM) (Figure 3A,D,G). Next, to obtain the corresponding ultrastructure of these cell components, the same sections were subsequently imaged using TEM (Figure 3B,E,H). All three structures were successfully aligned with high accuracy (Figure 3C,F,I). These results demonstrate that mScarlet-H is a high-performance red probe that can be generalized for same section CLEM imaging of various cellular targets.

### 3.4. Dual-Colour Post-Epon-Embedding Same Section CLEM Using mEosEM-E and mScarlet-H

The mScarlet-H and mEosEM-E were then combined to perform dual-colour same section CLEM imaging. The nuclear envelope (laminA) and mitochondria matrix in HEK 293T cells were labelled by mScarlet-H and mEosEM-E, respectively. After standard EM sample preparation, dual-colour wide-field FM images of cell sections (100 nm) were sequentially acquired with a 561-nm laser for 100 frames to image mScarlet-H, followed by 20 frames with a 488-nm laser, and 20 frames with 488- plus 405-nm lasers to image mEosEM-E. For the red channel, images were averaged to obtain the final image; while for the green channel, sLM was applied as described above. As shown in Figure 4A, the mScarlet-H-labelled laminA in the red channel exhibited a good nuclear envelope structure with a high SBR. However, in the green channel, regardless of whether mEosEM-E was in the on- (Figure 4B) or off-state (Figure 4C), the SBR of the fluorescence signal was very low. Furthermore, in addition to the mEosEM-E-labelled mitochondrial structure, the nuclear envelope structure due to mScarlet-H crosstalk could also be observed as indicated by the merged image of mEosEM-E (on-state) and mScarlet-H (Figure 4D, box). Notably, using the photoswitching property of mEosEM-E and the sCLEM method, the structure of mitochondria could be obtained with a very high SBR (Figure 4E). Moreover, the mScarlet-H crosstalk signal was effectively eliminated (Figure 4F) by the sCLEM imaging scheme, as the green signal of mScarlet-H does not possess the photoswitching property. A further investigation showed that the crosstalk phenomenon is universal for all the tested RFPs that survived standard EM sample preparation (Appendix A). A very likely reason is that under high-energy illumination, severe chromophore photoreduction happens that leads to obvious red-to-green photoconversion of the RPFs, a mechanism that has been reported before [25,26,27].

Next, we applied dual-colour CLEM to investigate how nucleolar proteins are organized in the nucleolus. The nucleolus of mammalian cells has three morphologically distinct components: the fibrillar centre (FC), the dense fibrillar component (DFC), and the granular component (GC), which perform different functions in ribosome biogenesis [28]. We labelled the GC marker B23 with mEosEM-E and labelled Nopp140, whose localization has been controversially reported to be at the FC or DFC [29,30], with mScarlet-H. The results showed that in the green channel, typical GC structures were clearly visible in the FM images after sLM, again exemplifying the utility of the subtraction strategy (Figure 4G–I). Merging of the green and red channels revealed that the red signal was encircled by the green signal (Figure 4J–L). Further correlative FM and EM images (Figure 4M–O) showed that the fluorescence signal of Nopp140 was closely matched to the solid high electron-dense regions (Figure 4N, arrowheads) in the EM image, indicating a DFC localization, which is consistent with the fact that it’s a functional different area other than GC (Figure 4M,O). Additionally, we also double-labelled Nop52 and Nopp140 by mEosEM-E and mScarlet-H, respectively. Unexpectedly, while Nop52 was often used as a marker of GC [31], Nopp140 was not located in the vacant holes devoid of the Nop52 signals, but partially overlapped with Nop52 (Appendix A). These results indicated that Nop52 localized both at the GC and the DFC.

Taken together, these results strongly demonstrate that mEosEM-E and mScarlet-H are optimal probes for high-quality dual-colour post-Epon-embedding CLEM imaging, and sCLEM with mEoEM-E offers a great advantage in reducing the green background of the Epon resin as well as in eliminating the signal crosstalk of RFPs.

## 4. Discussion

In current CLEM imaging, there is a trade-off between the quality of the FM and the EM data. Thus, it is challenging to obtain first-class FM and EM images of the same ultrathin section. Developing CLEM probes with resistance to the treatment of EM sample preparation is an efficient way to solve this problem, thereby, making CLEM with both high-quality FM and EM possible.

Very recently, several FPs were published for post-Epon-embedded CLEM. However, when these probes were used in practical CLEM imaging, the SBR of the FM images were very low. This problem is even more prominent for imaging low-abundance proteins. In the current study, we utilized the optical switching properties of RSFPs and proposed an sCLEM method, by which the FP signal can be easily distinguished from that of the unmodulated background. The sCLEM method works more efficiently than the OLID technique in our case, which may be due to the phenomenon that the background fluorescence also showed minor on and off responses to 405- and 488-nm excitation. Compared to the single-frame subtraction (Max-Min), subtraction between the multi-frame averages of the on and off states (SumAVG-SumAVG) has an additional benefit of largely attenuated noise. Notably, the sCLEM method hits two birds with one stone. In addition to effectively removing the background signal of the Epon resin and improving the SBR of the FM image, it also removes the signal crosstalk of the RFP in the green channel during dual-colour CLEM. To further improve the SBR of an FM image, we used mEosEM as the template and developed a variant, mEosEM-E, with enhanced on-state brightness and on/off contrast that is beneficial for fluorescent background elimination and weak signal extraction. We also identified mScarlet-H as the best RFP for post-Epon-embedding CLEM. This was validated by actual imaging applications, where we proved that mEosEM-E/mScarlet-H is an excellent FP pair for same section dual-colour CLEM of post-Epon-embedded samples. Compared with mEos4b, mEosEM retained higher fluorescence intensity after osmic acid treatment [16]. Considering that the fluorescence intensity retained in GMA-embedded samples is mainly dependent on the osmic acid-resistant properties of the FP, we suspected that mEosEM-E/mScarlet-H would also be excellent choices for the hydrophilic GMA resin.

Besides the development of an advanced green CLEM probe, we also performed a meticulous study that thoroughly dissected the influence of key chemical treatments during TEM sample preparation on the fluorescence preservation of RFPs. Interestingly, we found that the responses of RFPs to these treatments varied greatly. Some RFPs (mRuby3, tdTomato) showed good OsO_4_ resistance, however, their superiorities diminished after complete EM sample preparation, suggesting that they are more sensitive to the later sample treatment steps. Therefore, we reasoned that the OsO_4_ resistance assay is not a suitable gold standard for CLEM probe screening, even though OsO_4_ fixation seems to be the uppermost factor for fluorescence quenching. The development of superior CLEM probes requires investigating the probe’s performance after complete sample preparation, even though this will significantly increase the workload of probe development and add challenges for high-throughput screening. Furthermore, this study also provided useful information that may assist the rational design of new types of probes. For example, mRuby3 exhibited superior robustness towards dehydration, which may provide clues for developing FPs that can work in a hydrophobic environment.

The exact mechanism of fluorescence quenching of FPs during each step of EM sample preparation is unknown. Pre-fixation with PFA and GA may non-specifically cross link FPs and change their conformations, which affects their fluorescence [32]. The very strong oxidizing agent OsO4 can react with the surface amino acids of the barrel structure of FPs [14] and break peptide bonds, leading to quenching of FP fluorescence by oxidation [13,33]. Reducing the surface side-chain reactivity of FPs with fixation reagents may alleviate the fluorescence loss [14,16]. In addition, the low pH of the fixation buffer may partly turn FPs into protonated states with low fluorescence. However, fluorescence signals can be partially recovered by chemical reactivation using an alkaline buffer [33]. Dehydration with ethyl alcohol quenches FPs’ fluorescence, maybe because water is needed for FPs’ fluorescence as the relaxation of the excited state of GFP is related to a proton transfer chain that includes water [34]. Several issues may contribute to the loss of fluorescence during Epon-embedding. First, FPs may be cross linked with Epon causing a change in their conformation. Second, FPs might be partly or fully destroyed by the high temperature essential for the polymerization reaction of Epon. Third, the hydrophobic environment of the Epon polymer might also quench FP fluorescence, as under this condition, the absorbed energy of FPs is released to the polymer as thermal energy rather than emission of fluorescence photons [35]. In conclusion, besides low reactivity with the chemical reagents, high structural or conformational stability of FPs that is insensitive to harsh conditions during sample preparation is important for fluorescence preservation. The high resistance to the conventional EM sample preparation of mEosEM-E may be attributed to the low reactivity toward OsO4 and high chemical- and thermo-stability (same as mEosEM) that help survive the dehydration and high-temperature embedding. The most prominent advantage of mScarlet-H is its resistance to Epon embedding, which was most likely attributed to its high thermostability and insensitivity to the Epon resin.

Before this study, nucleolar proteins have never been inspected by CLEM on the same section. Our dual-colour CLEM results revealed that although both have been considered as a GC marker, Nop52 and B23 have different sub-nucleolar localizations. The partial co-localization of Nop52 with Nopp140 suggests that B23 would be a better marker to represent GC. Moreover, we confirmed a DFC localization of Nopp140 in our experimental conditions.

Further directions for the development of post-Epon-embedding CLEM probes would be: (1) the establishment of a high-throughput automatic on-section screening system; (2) a red RSFP CLEM probe for background-reduced and/or SR dual-colour CLEM, for which mScarlet-H may be a good starting template; (3) both green and red CLEM probes with even higher on-section brightness, SBR, and photostability.

## Figures and Tables

**Figure 1 cells-11-01077-f001:**
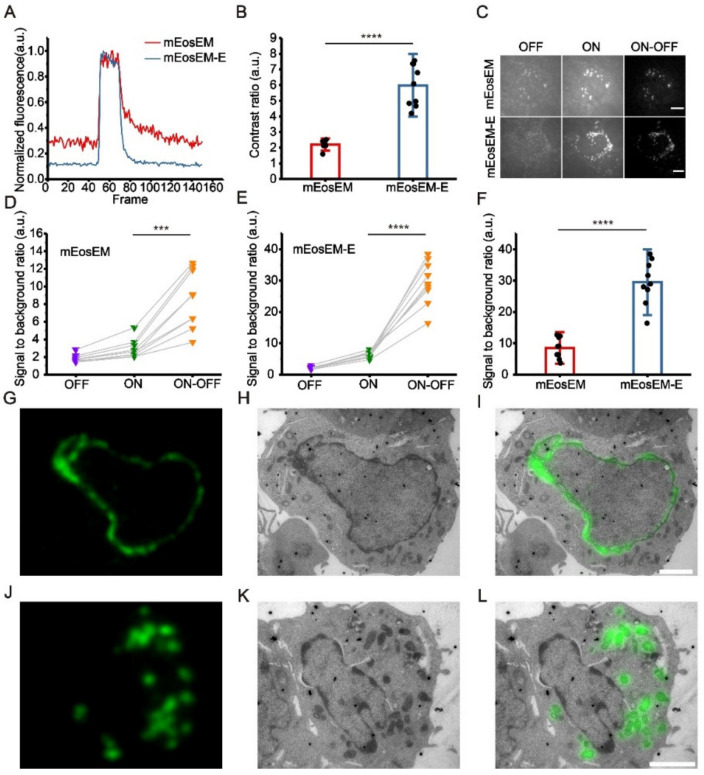
mEosEM-E is an improved green RSFP for high SBR sCLEM imaging. (**A**) Representative normalized photoswitching curves of Epon-embedded HEK 293T cell sections (100 nm) expressing mitochondria-targeted mEosEM (red) and mEosEM-E (blue). Samples were excited and switched off with the 488-nm laser (0.41 kW/cm^2^) and switched on with the 405-nm (0.21 kW/cm^2^) laser for 1 s. (**B**) Histogram of the photoswitching contrast ratio of mEosEM (red) and mEosEM-E (blue). Data are summarized in Appendix A. Bars represent mean ± SD. *p* values were determined with two-tailed *t*-test (*n* = 9). **** indicates *p* < 0.0001. (**C**) Representative OFF (left), ON (middle), and ON-OFF (right) images of Epon-embedded HEK 293T cell sections (100 nm) expressing mitochondria-targeted mEosEM and mEosEM-E. Scale bar, 5 µm. (**D**,**E**) SBR of mEosEM- (**D**) and mEosEM-E- (**E**) labelled cell samples in OFF (purple), ON (green), and ON-OFF (orange) images. Data are summarized in Appendix A and Appendix A. *p* values were determined with two-tailed *t*-tests (*n* = 9). *** indicates *p* < 0.001, **** indicates *p* < 0.0001. (**F**) Statistics SBR of mEosEM (red) and mEosEM-E (blue) in ON-OFF images. Two-tailed *t*-tests were performed (*n* = 9). **** indicates *p* < 0.0001. Data are summarized in Appendix A. (**G**–**L**) Post-Epon-embedding CLEM of mEosEM-E. FM, EM, and CLEM images of nuclear envelope (**G**–**I**) and mitochondria (**J**–**L**) labelled by mEosEM-E (100 nm slices). Scale bars, 2 µm.

**Figure 2 cells-11-01077-f002:**
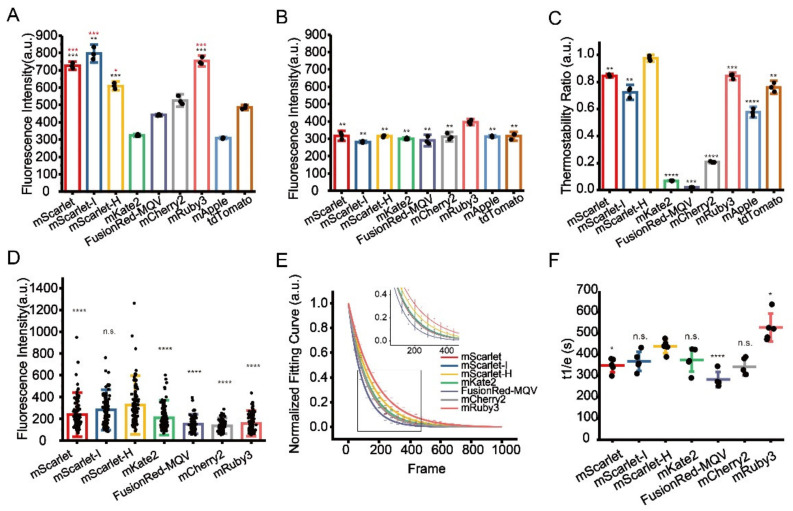
Identification of optimal RFPs for CLEM imaging. (**A**) Residual fluorescence intensity of RFPs after 1% OsO_4_ post-fixation. Red stars represent comparisons to mCherry2, black stars represent comparisons to mKate2. (**B**) Residual fluorescence intensity of RFPs after 1% OsO_4_ post-fixation followed by absolute ethanol dehydration. (**C**) Thermostability of RFPs at 60 °C. (**D**) Fluorescence intensity of RFPs on Epon-embedded sections (100 nm). (**E**) Normalized photobleaching curves of RFPs on Epon-embedded sections (100 nm). Enlarged view of the boxed area is shown on top. (**F**) Statistics of the photobleaching time when the fluorescence intensity of RFPs reduced to 1/e of their initials. Bars represent mean ± SD. *p* values were determined with two-tailed *t*-tests in (**A**–**C**) (*n* = 3) and (**F**) (*n* = 5), Mann–Whitney U test was performed in (**D**) (*n* = 106). n.s. indicates *p* > 0.05, * indicates *p* < 0.05, ** indicates *p* < 0.01, *** indicates *p* < 0.001, **** indicates *p* < 0.0001. Data are summarized in Appendix A, Appendix A, Appendix A, Appendix A and Appendix A.

**Figure 3 cells-11-01077-f003:**
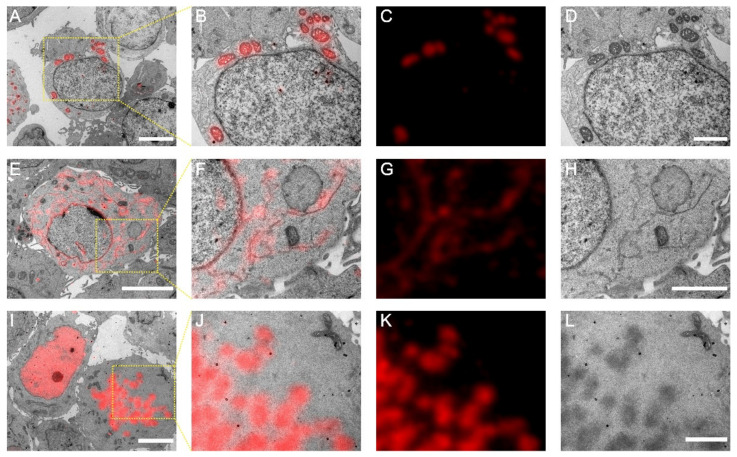
Post-Epon-embedding CLEM imaging with mScarlet-H. (**A**) CLEM image of the mitochondrial matrix. Scale bar, 5 µm. (**B**–**D**) Magnified CLEM, FM, and EM images of boxed area in (**A**). Scale bars, 2 µm. (**E**) CLEM image of the endoplasmic reticulum sec61β protein. Scale bar, 5 µm. (**F**–**H**) Magnified CLEM, FM, and EM images of boxed area in (**E**). Scale bar, 2 µm. (**I**) CLEM image of the nucleosome H2B protein. Scale bars, 5 µm. (**J**–**L**) Magnified CLEM, FM, and EM images of boxed area in (**I**). Scale bars, 2 µm.

**Figure 4 cells-11-01077-f004:**
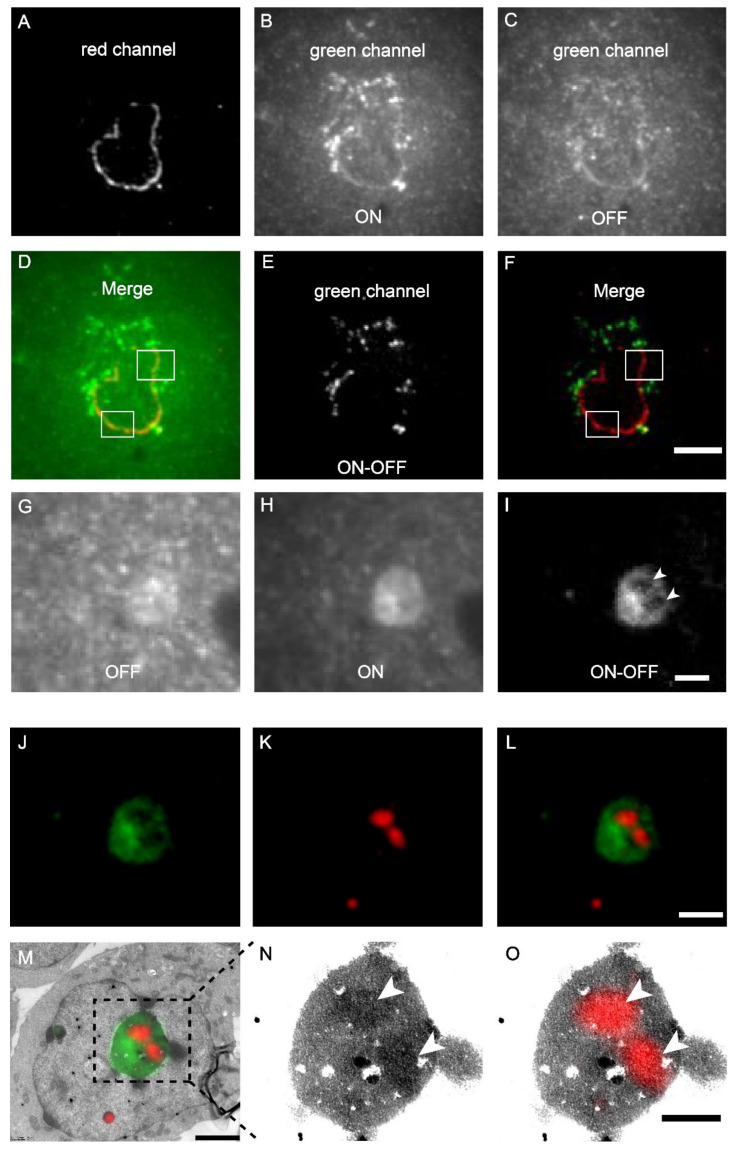
Dual-colour post-Epon-embedding CLEM using mEosEM-E and mScarlet-H. (**A**–**F**) Dual-colour imaging of mEosEM-E-labelled mitochondria and mScarlet-H-labelled LaminA protein in HEK 293T cell sections (100 nm). Representative red channel (**A**), green channel ON (**B**), and OFF (**C**) FM images. (**D**) Merged image of (**A**) and (**B**). White boxes indicate areas that showed crosstalk signals from mScarlet-H. (**E**) Green channel ON-OFF image. (**F**) Merged image of (**A**) and (**E**). White boxes indicate the same areas in (**D**). Scale bar, 5 µm. (**G**–**O**) Dual-colour CLEM imaging of nucleolar proteins. OFF (**G**), ON (**H**), and ON-OFF (**I**) images of mEosEM-E labelled B23 in HEK 293T cells sections (100 nm). White arrowheads indicate areas that were not visible in (**G**) and (**H**). Scale bar, 2 µm. Green channel FM (**J**), Red channel FM (**K**), merged channel FM (**L**), and CLEM (**M**) images of HEK 293T cell sections expressing mEosEM-E labelled B23 and mScarlet-H labelled Nopp140. Gamma value: 1.6 for both channels. Scale bars, 2 µm. (**N**–**O**) Enlarged EM (**N**) and CLEM (**O**) images of boxed area in (**M**), white arrowheads indicate the localization of Nopp140 that showed higher EM contrast compared to B23. Gamma value: 1.6 for red channel. Scale bar, 1 µm.

## Data Availability

Not applicable.

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
