# Peer review of "Improved Fluorescent Proteins for Dual-Colour Post-Embedding CLEM"

_cells, 2022, doi:10.3390/cells11071077_

Round 1
Reviewer 1 Report
very good paper, high quality and very useful results.
Comments:
- In Material and methods, the ultrathin sectioning is not explained. It would be good to include epon embedding and sectioning in the right place. It is not clear if is at 2.6 paragraph after dehydration because the fluorescence images are acquired on ultrathin sections on pioloform, or at the beginning of 2.9 paragraph because the high content screening is performed in the tissue after OsO4 and before Epon embedding.
Is the high content screening performed on tissue after osmium, epon blocks or ultrathin sections or... please clarify. - Since the comparison on the graphs, It would be also good to have an image of mEosEM to see the comparison with mEosEM-E in the imaging.
- It would be desirable, might be for the next paper, to have higher magnification EM images and appropiated contrast. For example, in the case of the nucleolus the differences between the components are not nicely seen, so the annotation of the fluorescence to one particular component is at that level of resolution quite risky.
Author Response
Point-by-point responses are provided below. The reviewers’ comments are shown in black, and our responses are highlighted in red.
Responses to Comments
Reviewer 1:
- In Material and methods, the ultrathin sectioning is not explained. It would be good to include epon embedding and sectioning in the right place. It is not clear if is at 2.6 paragraph after dehydration because the fluorescence images are acquired on ultrathin sections on pioloform, or at the beginning of 2.9 paragraph because the high content screening is performed in the tissue after OsO4 and before Epon embedding.
Is the high content screening performed on tissue after osmium, epon blocks or ultrathin sections or... please clarify.
Response: Thanks for the reviewer’s suggestion. The high-content screening was performed on cell samples with pre-fixation (Figure S5), pre-fixation + OsO4 post-fixation (Figure 2A), pre-fixation + OsO4 post-fixation + dehydration (Figure 2B), but without Epon embedding and sectioning. The high-content screenings were independent experiments different from that of CLEM imaging. For CLEM imaging, the “ultrathin sectioning” was described at the end of the paragraph “2.4 EM sample preparation”. Therefore, to make it clear, we have moved the paragraph “EM sample preparation” directly ahead of paragraph “2.9 CLEM imaging” in the revised manuscript.
- Since the comparison on the graphs, It would be also good to have an image of mEosEM to see the comparison with mEosEM-E in the imaging.
Response: We have added the required images of mEosEM to Figure 1C as shown below (Response figure 1C). We also have added corresponding descriptions in the manuscript (Line 276) and the figure legend (Line 298) as follows: “…ON - OFF image of mEosEM- and mEosEM-E-labeled cells…” and “…expressing mitochondria targeting mEosEM and mEosEM-E…”.
Response figure 1. mEosEM-E is an improved green RSFP for high SBR sCLEM imaging. (A) Representative normalized photoswitching curves of Epon-embedded HEK 293T cell sections (100 nm) expressing mitochondria-targeted mEosEM (red) and mEosEM-E (blue). Samples were excited and switched off with the 488-nm laser (0.41 kW/cm2), and switched on with the 405-nm (0.21 kW/cm2) laser for 1 s. (B) Statistics photoswitching contrast ratio of mEosEM (red) and mEosEM-E (blue). Bars represent mean ± SD. P values were determined with two-tailed t-test (n = 9). **** indicates p < 0.0001. (C) Representative OFF (left), ON (middle), and ON - OFF (right) images of Epon-embedded HEK 293T cell sections (100 nm) expressing mitochondria targeting mEosEM and mEosEM-E. Scale bar, 5 µm. (D, E) SBR of mEosEM- (D) and mEosEM-E- (E) labeled cell samples in OFF (purple), ON (green), and ON - OFF (orange) images. P values were determined with two-tailed t-tests (n = 9). *** indicates p < 0.001, **** indicates p < 0.0001. (F) Statistics SBR of mEosEM (red) and mEosEM-E (blue) in ON - OFF images. Two-tailed t-tests were performed (n = 9). **** indicates p < 0.0001. (G-L) Post-Epon-embedding CLEM of mEosEM-E. FM, EM, and CLEM images of nuclear envelope (G-I) and mitochondria (J-L) labeled by mEosEM-E (100 nm slices). Scale bars, 2 µm.
- It would be desirable, might be for the next paper, to have higher magnification EM images and appropiated contrast. For example, in the case of the nucleolus the differences between the components are not nicely seen, so the annotation of the fluorescence to one particular component is at that level of resolution quite risky.
Response: We apologize for having delivered the misleading message. The EM images shown in the paper were all acquired with high magnification (see Response table 1 below). When performing registration with FM image, the FM image (100 ×) was enlarged by interpolation to the same pixel size as that of the EM image. The original EM images were manually scaled down for image layout. We now have modified Figure 3 to better display the ultrastructure detail in the EM images as shown below (Response figure 2). We also have added a corresponding description in the figure legend of Figure 3 (Line 382-387). In Figure 4 N-O and Figure S9E-F, the EM images were already shown with high magnification (30, 000 × and 23, 000 ×). However, we did adjust the image contrast in Figure 4N-O and Figure S9E-F to highlight the electron-dense regions in the nucleolus, otherwise, the differences were hard to discriminate in the original EM images (Response figure 3). In the future, we will optimize the protocol of the EM sample preparation to enhance the contrast differences of the nucleolus structures under EM.
|
Fig. 1I |
Fig. 1L |
Fig. 2A |
Fig. 2D |
Fig. 2E |
Fig. 2H |
Fig. 2I |
Fig. 2L |
Fig. 4M |
Fig. 4O |
Fig. S9C |
Fig. S9C |
|
|
Magnification |
9300 × |
13000 × |
4800 × |
11000 × |
6800 × |
18500 × |
4800 × |
11000 × |
11000 × |
30000 × |
9300 × |
23000 × |
Response table 1. Magnification of EM images.
Response figure 2. Post-Epon-embedding CLEM imaging by mScarlet-H. (A) CLEM image of the mitochondrial matrix. Scale bar, 5 µm. (B-D) Magnified CLEM, FM, and EM images of boxed area in (A). Scale bars, 2 µm. (E) CLEM image of the endoplasmic reticulum sec61β protein. Scale bar, 5 µm. (F-H) Magnified CLEM, FM, and EM images of boxed area in (E). Scale bar, 2 µm. (I) CLEM image of the nucleosome H2B protein. Scale bars, 5 µm. (J-L) Magnified CLEM, FM, and EM images of boxed area in (I). Scale bars, 2 µm.
Response figure 3. CLEM and EM images of mScarlet-H-labeled Nopp140. (A, D) CLEM images of mScarlet-H-labeled Nopp140. (B, E) Contrast-adjusted EM images of (A, D). (C, F) Original EM images of (A, D).
Reviewer 2 Report
This is an interesting manuscript describing significant improvement of application of fluorescent proteins for correlative light and electron microscopy (CLEM). Authors created a new mutant, mEosFP-EM-E, that possesses better stability of its green fluorescence under conditions of sample preparation for EM. Moreover, it was demonstrated that reversible ON-OFF photoswitching of mEosFP-EM-E can be used to strongly enhance signal-to-background ratio. In addition, several known red fluorescent proteins were tested for their use in samples for EM; mScarlet-H was selected as the best RFP for CLEM. Finally, dual-color CLEM with mEosFP-EM-E plus mScarlet-H was performed. Overall, this work provides significant technical advance and will probably be of interest for a wide audience of researchers. At the same time, I think that the manuscript can be improved in the following (minor) points:
- RFPs fluorescence changes at different steps are shown in a number of panels and tables. It would be helpful to combine these data into a single entity (plot, histogram, or table) with percentage of fluorescence loss throughout the EM sample preparation procedure(i.e., 100% for unfixed cells, then e.g. 50% for fixed cell, then e.g. 15% for dehydrated cells, etc).
- EM-CCD camera was used for fluorescence images acquisition. It is highly sensitive, down to single molecule detection. Is it really important to use EM-CCD or less sensitive cameras can also be used? Please comment. Also, detection gain should be noted in the Methods.
- The paper contains EM images of relatively low magnification comparable to that of light microscopy. Is it possible to image the obtained samples at high magnification to visualize the ultrastructure?
Author Response
Point-by-point responses are provided below. The reviewers’ comments are shown in black, and our responses are highlighted in red.
Responses to Comments
Reviewer 2:
This is an interesting manuscript describing significant improvement of application of fluorescent proteins for correlative light and electron microscopy (CLEM). Authors created a new mutant, mEosFP-EM-E, that possesses better stability of its green fluorescence under conditions of sample preparation for EM. Moreover, it was demonstrated that reversible ON-OFF photoswitching of mEosFP-EM-E can be used to strongly enhance signal-to-background ratio. In addition, several known red fluorescent proteins were tested for their use in samples for EM; mScarlet-H was selected as the best RFP for CLEM. Finally, dual-color CLEM with mEosFP-EM-E plus mScarlet-H was performed. Overall, this work provides significant technical advance and will probably be of interest for a wide audience of researchers. At the same time, I think that the manuscript can be improved in the following (minor) points:
- RFPs fluorescence changes at different steps are shown in a number of panels and tables. It would be helpful to combine these data into a single entity (plot, histogram, or table) with percentage of fluorescence loss throughout the EM sample preparation procedure (i.e., 100% for unfixed cells, then e.g. 50% for fixed cell, then e.g. 15% for dehydrated cells, etc).
Response: We agreed that it would be very helpful to show and compare all the data in one entity, assuming all the data were acquired with the same batch of samples and under the same imaging condition. However, practically, it is quite hard to do so because the treatments such as OsO4 post-fixation and Epon-embedding severely deteriorate the fluorescence so the imaging condition used in the earlier step will not work in the later step. While prolonged exposure time will enhance the weak signals in the later steps, it will saturate the signals in the earlier steps. Thus, for Figure 2A and 2B, we used the same imaging condition, but for Figure 2D, we changed the imaging condition. For each panel and comparison of different fluorescent proteins under the same sample preparation conditions, we used the same exposure time. As suggested by the reviewer, we modified the vertical scale in Figure 2A and B to the same level for easier exhibiting the quantitative differences as shown below (Response figure 4A, B).
Response figure 4. Identification of optimal RFPs for CLEM imaging. (A) Residual fluorescence intensity of RFPs after 1% OsO4 post-fixation. Red stars represent comparisons to mCherry2, black stars represent comparisons to mKate2. (B) Residual fluorescence intensity of RFPs after 1% OsO4 post-fixation followed by absolute ethanol dehydration. (C) Thermostability of RFPs at 60 °C. (D) Fluorescence intensity of RFPs on Epon-embedded sections (100 nm). (E) Normalized photobleaching curves of RFPs on Epon-embedded sections (100 nm). Enlarged view of the boxed area is shown on top. (F) Statistics of the photobleaching time when the fluorescence intensity of RFPs reduced to 1/e of their initials. Bars represent mean ± SD. P values were determined with two-tailed t-tests in (A-C) (n = 3) and (F) (n = 5), Mann-Whitney U test was performed in (D) (n = 106). n.s. indicates p > 0.05, * indicates p < 0.05, ** indicates p < 0.01, *** indicates p < 0.001, **** indicates p < 0.0001.
- EM-CCD camera was used for fluorescence images acquisition. It is highly sensitive, down to single molecule detection. Is it really important to use EM-CCD or less sensitive cameras can also be used? Please comment. Also, detection gain should be noted in the Methods.
Response: We have tested that the sCMOS camera could also be used for image acquisition, however, the performance was inferior to that acquired by EM-CCD, because the fluorescence signals were still relatively weak after EM sample preparation compared to that of the conventional FM. The detection gain was 300, which has been noted as suggested in the Methods (2.9).
- The paper contains EM images of relatively low magnification comparable to that of light microscopy. Is it possible to image the obtained samples at high magnification to visualize the ultrastructure?
Response: We apologize for having delivered the misleading message. The EM images shown in the paper were all acquired with high magnification (see Response table 1 below). When performing registration with FM image, the FM image (100 ×) was enlarged by interpolation to the same pixel size as that of the EM image. The original EM images were manually scaled down for image layout. We now have modified Figure 3 to better display the ultrastructure detail in the EM images as shown below (Response figure 2). We also have added a corresponding description in the figure legend of Figure 3 (Line 382-387). In Figure 4 N-O and Figure S9E-F, the EM images were already shown with high magnification (30, 000 × and 23, 000 ×). However, we did adjust the image contrast in Figure 4N-O and Figure S9E-F to highlight the electron-dense regions in the nucleolus, otherwise, the differences were hard to discriminate in the original EM images (Response figure 3). In the future, we will optimize the protocol of the EM sample preparation to enhance the contrast differences of the nucleolus structures under EM.
|
Fig. 1I |
Fig. 1L |
Fig. 2A |
Fig. 2D |
Fig. 2E |
Fig. 2H |
Fig. 2I |
Fig. 2L |
Fig. 4M |
Fig. 4O |
Fig. S9C |
Fig. S9C |
|
|
Magnification |
9300 × |
13000 × |
4800 × |
11000 × |
6800 × |
18500 × |
4800 × |
11000 × |
11000 × |
30000 × |
9300 × |
23000 × |
Response table 1. Magnification of EM images.
Response figure 2. Post-Epon-embedding CLEM imaging by mScarlet-H. (A) CLEM image of the mitochondrial matrix. Scale bar, 5 µm. (B-D) Magnified CLEM, FM, and EM images of boxed area in (A). Scale bars, 2 µm. (E) CLEM image of the endoplasmic reticulum sec61β protein. Scale bar, 5 µm. (F-H) Magnified CLEM, FM, and EM images of boxed area in (E). Scale bar, 2 µm. (I) CLEM image of the nucleosome H2B protein. Scale bars, 5 µm. (J-L) Magnified CLEM, FM, and EM images of boxed area in (I). Scale bars, 2 µm.
Response figure 3. CLEM and EM images of mScarlet-H-labeled Nopp140. (A, D) CLEM images of mScarlet-H-labeled Nopp140. (B, E) Contrast-adjusted EM images of (A, D). (C, F) Original EM images of (A, D).
Reviewer 3 Report
In this paper, the authors report an improved mutated version (H63E) of their previously reported photoswitchable (PS) green fluorescent protein (GFP), mEosEM, developed for use in correlative light and electron microscopy (CLEM), and demonstrate the utility of mEosEM-E for two-color CLEM when paired with a red fluorescent protein (RFP), performing a thorough characterization and evaluation of potential partner RFPs before arriving on mScarlet-H as the optimal choice. Additionally, the authors take advantage of the PS properties of mEosEM to significantly improve its signal to background ratio (SBR) in Epon resin-embedded samples, where autofluorescence from the resin can be a complicating factor, and reduce cross-talk. Finally, the authors show that their method and choice of dyes can be used to unambiguously resolve an outstanding question in the literature, namely the location of Nopp140 (fibrillary center, FC, or dense fibrillary component, DFC).
The manuscript is well written, and the experiments appear thorough, well designed, carefully performed, and described in detail. Furthermore, the paper makes a significant contribution to improving CLEM. In the opinion of this reviewer, only minor revisions are required before publishing this excellent paper. My main request would be to add more discussion and insight/speculation into the underlying mechanism(s) of fluorescence quenching/loss and why the authors believe the two particular proteins identified, mEosEM-E and mScarlet-H, perform better than the alternatives studied, as this could provide direction for further optimization/improvement (either through modifying the fluorescent proteins or CLEM sample preparation). More detailed comments follow:
- Major questions:
- What portion(s) of the “chemical treatments” for EM sample preparation (lines 52-53) cause(s) the quenching of fluorescent proteins (lines 72-73, 98, 102-104, and elsewhere) and how/why - i.e., what is the mechanism, and how might it be circumvented? This would be nice to have addressed, with references as appropriate.
- I see this is later partially addressed in Sxn 3.2, through carefully measuring the fluorescence of the RFPs after each EM sample preparation step (e.g., lines 298-301), which is greatly appreciated, but more discussion as to the proposed mechanism of fluorescence quenching/degradation (not just providing the relative performance of the various RFPs at each step in the process) would be nice.
- In Fig S5, it might be more beneficial to use the same vertical scale for Panels A and B to more readily indicate the degree of fluorescence loss upon pre-fixation. Likewise, using the same scale in Fig S6 (while stretching the length of the Y axis so that the reduced fluorescence values can still be read off and differences among the RFPs observed) would be useful to the reader.
- Likewise, employing the same vertical scales in Fig 2a, b, and d would aid direct comparison, especially since the fluorescence intensities in panels b and d appear to be somewhat similar, but significantly lower than in panel a (i.e., before dehydration).
- Line 317 - fluorescence of mScarlet-H and -I doesn't appear to differ by a statistically significant amount (Fig S7), as the uncertainty bars overlap.
- While the fluorescence intensity cross-sectional profile in Panel B of Fig S1 is very useful for showing the level of background fluorescence from an Epon slice, a color version of the image in Panel A in Fig S1 showing the green background fluorescence would be nice if available.
- Line 121 - an immediate question that arises is what mutation(s) are present in mEosEM-E (and is there any insight into why they are beneficial in terms of performance as a FP for CLEM)? This is later addressed later (line 244), but the identity of the exact mutation could perhaps be better introduced earlier at line 121.
- How much better is -E than mutant versions -A, -I, -L, and -V (line 244)?
- Hard to tell just from Fig S2 since the plotted flourescence intensities are all normalized. For example, how do their SBRs compare with each other and the base mEosEM?
- Statistical tables comparing the relative SBR performance of these mutants (similar to those in the Supplementary Materials for the RFPs or comparing mEosEM to mEosEM-E) should be added.
- I presume the mutations in each of mEosEM-A through -Y are at H63 (line 244) and their precise identities are given by the corresponding single letter amino acid codes, but it would help if this were explicitly stated. In addition, based on this, similar to with the RFPs, more discussion of the proposed mechanism underlying improved performance would be appreciated (particularly for -A, -I, -L, and -V in addition to -E).
- Minor typos
- Lines 26-27 - mScarlet-H is misspelled as mScharlet-H in the abstract.
- Line 56, 60, 65, 87, 96, 103, 104, 152, 177, 178, 292, 294, 302, 309, 310, 323, 334, 335, 456, 458, 459, 489, 490 - subscript needed for OsO4
- Line 69 - mCherry2 (consistency in capitalization)
- Line 143 - subscript needed for CO2
- Lines 163 and 164 - superscript needed for cm2
- Line 367 - typo (wide-field, not wide-filed)
- Line 421 - should read “boxed area in (M)” (missing “M”).
Author Response
Point-by-point responses are provided below. The reviewers’ comments are shown in black, and our responses are highlighted in red.
Responses to Comments
Reviewer 3:
In this paper, the authors report an improved mutated version (H63E) of their previously reported photoswitchable (PS) green fluorescent protein (GFP), mEosEM, developed for use in correlative light and electron microscopy (CLEM), and demonstrate the utility of mEosEM-E for two-color CLEM when paired with a red fluorescent protein (RFP), performing a thorough characterization and evaluation of potential partner RFPs before arriving on mScarlet-H as the optimal choice. Additionally, the authors take advantage of the PS properties of mEosEM to significantly improve its signal to background ratio (SBR) in Epon resin-embedded samples, where autofluorescence from the resin can be a complicating factor, and reduce cross-talk. Finally, the authors show that their method and choice of dyes can be used to unambiguously resolve an outstanding question in the literature, namely the location of Nopp140 (fibrillary center, FC, or dense fibrillary component, DFC).
The manuscript is well written, and the experiments appear thorough, well designed, carefully performed, and described in detail. Furthermore, the paper makes a significant contribution to improving CLEM. In the opinion of this reviewer, only minor revisions are required before publishing this excellent paper. My main request would be to add more discussion and insight/speculation into the underlying mechanism(s) of fluorescence quenching/loss and why the authors believe the two particular proteins identified, mEosEM-E and mScarlet-H, perform better than the alternatives studied, as this could provide direction for further optimization/improvement (either through modifying the fluorescent proteins or CLEM sample preparation). More detailed comments follow:
Response: We thank the reviewer for the very constructive suggestion. A detailed discussion regarding the underlying mechanisms of fluorescence loss during each step of EM sample preparation and the possible reasons that mEosEM-E and mScarlet-H performed better than the alternatives are given in the next response paragraph.
Major questions:
- What portion(s) of the “chemical treatments” for EM sample preparation (lines 52-53) cause(s) the quenching of fluorescent proteins (lines 72-73, 98, 102-104, and elsewhere) and how/why - i.e., what is the mechanism, and how might it be circumvented? This would be nice to have addressed, with references as appropriate.
- I see this is later partially addressed in Sxn 3.2, through carefully measuring the fluorescence of the RFPs after each EM sample preparation step (e.g., lines 298-301), which is greatly appreciated, but more discussion as to the proposed mechanism of fluorescence quenching/degradation (not just providing the relative performance of the various RFPs at each step in the process) would be nice.
Response: The reviewer raised an important question that we are also very interested in. However, we have no answers since no direct experimental evidence was available. As suggested by the reviewer, we may get some clues from the literature. Therefore, we have added a discussion in the manuscript (Line 497-516) regarding the mechanism of fluorescence quenching during EM sample preparation and how it might be circumvented as follows: “The exact mechanism of fluorescence quenching of FPs during each step of EM sample preparation is unknown. Pre-fixation with PFA and GA may nonspecifically cross link FPs and change their conformations, which affects their fluorescence [32]. OsO4 is a very strong oxidizing agent that can react with the surface amino acids of barrel structure of FPs [14] and break peptide bonds, and thus the FPs’ fluorescence can be largely quenched by oxidation [13, 33]. Reducing the surface side-chain reactivity of FPs with fixation reagents may alleviate the fluorescence loss [14, 16]. In addition, the low pH of the fixation buffer also partly turns FPs into protonated states with low fluorescence. However, fluorescence signals can be partially recovered by chemical reactivation using alkaline buffer [33]. Dehydration with ethyl alcohol quenches FPs’ fluorescence, may because water is needed for FPs’ fluorescence since the relaxation of the excited state of GFP is related to a proton transfer chain that includes water [34]. Several issues may contribute to the loss of fluorescence during Epon-embedding. First, FPs may be cross linked with Epon and changed their conformation. Second, FPs might be partly or fully destroyed by the high temperature essential for the polymerization reaction of Epon. Third, the hydrophobic environment of Epon polymer might also quench FP fluorescence, may because under this condition, the absorbed energy of FPs is released to the polymer as thermal energy rather than emits as fluorescence photons [35]. In conclusion, besides low reactivity with the chemical reagents, high structural or conformational stability of FPs that is insensitive to harsh conditions during sample preparation is important for fluorescence preservation.” We have also given possible explanations that why our probes mEosEM/mScarlet-H can perform better than others in the manuscript (Line 516-521) as follows: “The high resistance to the conventional EM sample preparation of mEosEM-E may be attributed to the low reactivity toward OsO4 and high chemical- and thermo-stability (same as mEosEM) that help survive the dehydration and high-temperature embedding. The most prominent advantage of mScarlet-H is its resistance to Epon embedding, which was most likely attributed to its high thermostability and insensitivity to Epon resin.”
- In Fig S5, it might be more beneficial to use the same vertical scale for Panels A and B to more readily indicate the degree of fluorescence loss upon pre-fixation. Likewise, using the same scale in Fig S6 (while stretching the length of the Y axis so that the reduced fluorescence values can still be read off and differences among the RFPs observed) would be useful to the reader.
Response: As the reviewer suggested, we have used the same vertical scale of Panels A and B in Fig S5 as shown below (Response figure 5), but not in Fig S6. This is because, while the data of Panels A and B in Fig S5 were acquired with the same sample and under the same imaging condition, the data in Fig S6 were acquired with a different batch of samples, so the data in Fig S6 could not be used to indicate the degree of fluorescence loss.
Response figure 5. Pre-fixation influence on RFPs. (A) Fluorescence intensity of RFPs in live cells. (B) Fluorescence intensity of RFPs in pre-fixed cells.
- Likewise, employing the same vertical scales in Fig 2a, b, and d would aid direct comparison, especially since the fluorescence intensities in panels b and d appear to be somewhat similar, but significantly lower than in panel a (i.e., before dehydration).
Response: We agreed that it would be very helpful to show and compare all the data in one entity, assuming all the data were acquired with the same batch of samples and under the same imaging condition. However, practically, it is quite hard to do so because the treatments such as OsO4 post-fixation and Epon-embedding severely deteriorate the fluorescence so the imaging condition used in the earlier step will not work in the later step. While prolonged exposure time will enhance the weak signals in the later steps, it will saturate the signals in the earlier steps. Thus, for Figure 2A and 2B, we used the same imaging condition, but for Figure 2D, we changed the imaging condition. For each panel and comparison of different fluorescent proteins under the same sample preparation conditions, we used the same exposure time. As suggested by the reviewer, we modified the vertical scale in Figure 2A and B to the same level for easier exhibiting the quantitative differences as shown below (Response figure 4A, B).
Response figure 4. Identification of optimal RFPs for CLEM imaging. (A) Residual fluorescence intensity of RFPs after 1% OsO4 post-fixation. Red stars represent comparisons to mCherry2, black stars represent comparisons to mKate2. (B) Residual fluorescence intensity of RFPs after 1% OsO4 post-fixation followed by absolute ethanol dehydration. (C) Thermostability of RFPs at 60 °C. (D) Fluorescence intensity of RFPs on Epon-embedded sections (100 nm). (E) Normalized photobleaching curves of RFPs on Epon-embedded sections (100 nm). Enlarged view of the boxed area is shown on top. (F) Statistics of the photobleaching time when the fluorescence intensity of RFPs reduced to 1/e of their initials. Bars represent mean ± SD. P values were determined with two-tailed t-tests in (A-C) (n = 3) and (F) (n = 5), Mann-Whitney U test was performed in (D) (n = 106). n.s. indicates p > 0.05, * indicates p < 0.05, ** indicates p < 0.01, *** indicates p < 0.001, **** indicates p < 0.0001.
- Line 317 - fluorescence of mScarlet-H and -I doesn't appear to differ by a statistically significant amount (Fig S7), as the uncertainty bars overlap.
Response: We apologize for the mistake. Yes, there is no significant difference between mScarlet-H and mScarlet-I in terms of in-resin fluorescence and SBR. We have changed the wording in the manuscript “mScarlet-H, which preserved the highest on-section fluorescence and SBR among the nine RFPs tested, is the best RFP reported to date for the same section post-Epon-embedding CLEM.” to be “Using this approach, we found that mScarlet-H exhibited higher brightness and SBR in the Epon resin compared with previously reported RFPs, and is suitable for same section post-Epon-embedding CLEM.” (Line 107); “mScarlet-H showed the highest residue fluorescence, followed by mScarlet-I, which are both significantly higher than the previously reported mKate2 and mCherry2” to be “mScarlet-H and mScarlet-I both showed the highest residue fluorescence, which is significantly higher than those of previously reported mKate2 and mCherry2” (Line 335-336); “Statistic results showed that mScarlet-H has the highest SBR among tested RFPs” to be “Statistic results showed that mScarlet-H and mScarlet-I have the highest SBR among the tested RFPs”. (Line337-338) However, it will not change the conclusion that mScarlet-H has the best overall performance as it has a higher average photostability than mScarlet-I.
- While the fluorescence intensity cross-sectional profile in Panel B of Fig S1 is very useful for showing the level of background fluorescence from an Epon slice, a color version of the image in Panel A in Fig S1 showing the green background fluorescence would be nice if available.
Response: The background fluorescence in Panel A in Fig S1 has been colored green as suggested by the reviewer.
- Line 121 - an immediate question that arises is what mutation(s) are present in mEosEM-E (and is there any insight into why they are beneficial in terms of performance as a FP for CLEM)? This is later addressed later (line 244), but the identity of the exact mutation could perhaps be better introduced earlier at line 121.
Response: We have changed the expression as “mEosEM-E (mEosEM H63E)” at Line 124.
- How much better is -E than mutant versions -A, -I, -L, and -V (line 244)?
Response: We have added a histogram and a statistical table as shown below (Response figure 6 and Response table 2) to quantitatively compare the on/off contrast ratios of mEosEM-E and the mutant versions -A, -L, and -V after EM sample preparation. However, we didn’t include mEosEM-I since this mutant showed relatively low brightness even before EM sample preparation (Response figure 7) and thus was excluded for further screening.
Response figure 6. Normalized contrast ratio comparison of mEosEM-A, mEosEM-E, mEosEM-L, and mEosEM-V after EM sample preparation. Bars represent mean ± SD. P-value were determined with two-tailed t-test in (n = 9). ** indicates p<0.01, *** indicates p<0.001, **** indicates p<0.0001
|
mEosEM-A |
mEosEM-E |
mEosEM-L |
mEosEM-V |
|
|
Normalized |
0.7898 |
0.7795 |
0.4608 |
0.2846 |
|
contrast |
0.5904 |
1.2324 |
0.5736 |
0.3591 |
|
ratio |
0.6438 |
1.2649 |
0.7698 |
0.4433 |
|
0.6925 |
1.2298 |
0.6045 |
0.4838 |
|
|
0.7208 |
0.8296 |
0.4409 |
0.4235 |
|
|
0.7013 |
0.7010 |
0.4461 |
0.5185 |
|
|
0.8798 |
0.8083 |
0.8770 |
0.6377 |
|
|
0.5533 |
1.1361 |
0.7154 |
0.5568 |
|
|
0.6208 |
1.0176 |
0.8364 |
0.3684 |
|
|
mean |
6.1930 |
8.9997 |
5.7251 |
4.0761 |
|
s.d. |
0.1015 |
0.2234 |
0.1703 |
0.1092 |
|
P value |
0.0028 |
0.0014 |
2.9832E-05 |
Response table 2. Statistics for a normalized contrast ratio of mEosEM-A, mEosEM-E, mEosEM-L, and mEosEM-V after EM sample preparation. Two-tailed t-tests were performed between mEosEM-E and other mutants, n = 9.
Response figure 7. Brightness screen of mEosEM and mutants in E. coli. before EM sample preparation.
- Hard to tell just from Fig S2 since the plotted flourescence intensities are all normalized. For example, how do their SBRs compare with each other and the base mEosEM?
Response: We have added a histogram in Figure S2 to show the quantitative comparison of the on/off contrast ratio of these mutants and mEosEM before EM sample preparation as shown below (Response figure 8).
Response figure 8. Photoswitching properties of mEosEM and mutants before EM sample preparation. (A-Q) Normalized fluorescence intensity of mEosEM and mutants was plotted against time. Cells expressing different fluorescent proteins were continuously illuminated with a 488-nm laser (8 W/cm2), while every 10 s, a 405-nm laser pulse (0.1 s, 9 W/cm2) were applied to switch on the FPs. Exposure time, 50 ms. (R) Mean contrast ratios of mEosEM and mutants before EM sample preparation. Error bars represent standard errors. n = 5.
- Statistical tables comparing the relative SBR performance of these mutants (similar to those in the Supplementary Materials for the RFPs or comparing mEosEM to mEosEM-E) should be added.
Response: We have added a statistical table (Table S10) in the Supplementary Material as shown in Response table 3 to compare the performance of these mutants before EM sample preparation as shown below.
|
Contrast ratio |
mean |
s.d. |
|||||
|
mEosEM |
0.623 |
1.114 |
0.845 |
0.821 |
1.673 |
1.015 |
0.407 |
|
mEosEM-A |
20.716 |
32.975 |
12.130 |
51.253 |
27.805 |
28.976 |
14.711 |
|
mEosEM-C |
0.840 |
1.117 |
4.340 |
6.428 |
1.744 |
2.894 |
2.412 |
|
mEosEM-D |
0.758 |
0.814 |
0.408 |
0.751 |
0.577 |
0.662 |
0.167 |
|
mEosEM-E |
39.382 |
32.132 |
47.701 |
24.849 |
11.781 |
31.169 |
13.762 |
|
mEosEM-F |
1.428 |
0.626 |
0.691 |
0.505 |
0.488 |
0.747 |
0.389 |
|
mEosEM-G |
15.291 |
19.369 |
14.976 |
34.559 |
13.255 |
19.490 |
8.718 |
|
mEosEM-I |
102.815 |
62.832 |
81.008 |
33.619 |
95.741 |
75.203 |
27.840 |
|
mEosEM-K |
2.102 |
1.887 |
1.420 |
1.964 |
1.854 |
1.845 |
0.256 |
|
mEosEM-L |
62.654 |
17.426 |
88.161 |
55.673 |
35.340 |
51.851 |
26.960 |
|
mEosEM-M |
1.926 |
1.776 |
2.145 |
2.118 |
2.663 |
2.126 |
0.336 |
|
mEosEM-N |
3.957 |
4.793 |
4.574 |
2.872 |
5.173 |
4.274 |
0.899 |
|
mEosEM-Q |
22.014 |
29.660 |
16.040 |
46.654 |
17.260 |
26.326 |
12.559 |
|
mEosEM-S |
4.546 |
8.633 |
13.426 |
10.471 |
15.886 |
10.592 |
4.372 |
|
mEosEM-T |
40.195 |
44.538 |
58.869 |
17.686 |
26.349 |
37.527 |
16.050 |
|
mEosEM-V |
228.326 |
73.234 |
174.318 |
188.631 |
167.465 |
166.395 |
57.168 |
|
mEosEM-Y |
0.757 |
1.103 |
1.248 |
1.331 |
1.028 |
1.093 |
0.223 |
Response table 3. Statistics for contrast ratio of mEosEM and mutants before EM sample preparation. n = 5.
- I presume the mutations in each of mEosEM-A through -Y are at H63 (line 244) and their precise identities are given by the corresponding single letter amino acid codes, but it would help if this were explicitly stated. In addition, based on this, similar to with the RFPs, more discussion of the proposed mechanism underlying improved performance would be appreciated (particularly for -A, -I, -L, and -V in addition to -E).
Response: We renamed the mutants in Figure S2 as mEosEM-H63X to clarify their identities. We are also interested in the underlying mechanism of the improved contrasts of some of the mutants. However, we noticed that amino acids with different features (A, L, and V are all hydrophobic, and E is negatively charged) have the same effects, which is quite confusing. Further investigation is needed. Time-resolved crystallography and transient absorption spectroscopy could be helpful.
Minor typos
- Lines 26-27 - mScarlet-H is misspelled as mScharlet-H in the abstract.
- Line 56, 60, 65, 87, 96, 103, 104, 152, 177, 178, 292, 294, 302, 309, 310, 323, 334, 335, 456, 458, 459, 489, 490 - subscript needed for OsO4
- Line 69 - mCherry2 (consistency in capitalization)
- Line 143 - subscript needed for CO2
- Lines 163 and 164 - superscript needed for cm2
- Line 367 - typo (wide-field, not wide-filed)
- Line 421 - should read “boxed area in (M)” (missing “M”).
Response: We apologized for these typos. They have all been corrected in the revised MS.